# Structural and Thermal Analysis of Softwood Lignins from a Pressurized Hot Water Extraction Biorefinery Process and Modified Derivatives

**DOI:** 10.3390/molecules24020335

**Published:** 2019-01-18

**Authors:** Lucas Lagerquist, Andrey Pranovich, Ivan Sumerskii, Sebastian von Schoultz, Lari Vähäsalo, Stefan Willför, Patrik Eklund

**Affiliations:** 1Johan Gadolin Process Chemistry Centre, Åbo Akademi University, Biskopsgatan 8, 20500 Turku/Åbo, Finland; llagerqu@abo.fi (L.L.); apranovi@abo.fi (A.P.); swillfor@abo.fi (S.W.); 2Department of Chemistry, Saint Petersburg State Forest Technical University, 194021 Saint Petersburg, Russia; 3Division of Chemistry of Renewable Resources, Department of Chemistry, University of Natural Resources and Life Sciences, Konrad-Lorenz-Strasse 24, A-3430 Tulln, Austria; ivan.sumerskii@boku.ac.at; 4CH-Bioforce Oy, Ahventie 4 A 21-22, FIN-02170 Espoo, Finland; Sebastian.schoultz@ch-bioforce.com (S.v.S.); lari.vahasalo@ch-bioforce.com (L.V.)

**Keywords:** lignin characterization, pressurized hot water extraction, biorefinery, condensed structures, NMR analysis, thermal analysis, lignin modification

## Abstract

In this work we have analyzed the pine and spruce softwood lignin fraction recovered from a novel pressurized hot water extraction pilot process. The lignin structure was characterized using multiple NMR techniques and the thermal properties were analyzed using thermal gravimetric analysis. Acetylated and selectively methylated derivatives were prepared, and their structure and properties were analyzed and compared to the unmodified lignin. The lignin had relatively high molar weight and low PDI values and even less polydisperse fractions could be obtained by fractionation based on solubility in *i*-PrOH. Condensation, especially at the 5-position, was detected in this sulphur-free technical lignin, which had been enriched with carbon compared to the milled wood lignin (MWL) sample of the same wood chips. An increase in phenolic and carboxylic groups was also detected, which makes the lignin accessible to chemical modification. The lignin was determined to be thermally stable up to (273–302 °C) based on its T_dst 95%_ value. Due to the thermal stability, low polydispersity, and possibility to tailor its chemical properties by modification of its hydroxyl groups, possible application areas for the lignin could be in polymeric blends, composites or in resins.

## 1. Introduction

As there is an uncertainty surrounding fossil-based raw materials, new alternative sources need to be explored for fuel and platform chemicals. Lignin is the most abundant aromatic biopolymer and could potentially be such an alternative source. The structure of lignin, in its native form, consists of phenylpropanoic units bonded together mainly by alkyl-aryl ether bonds that are formed by radical coupling reactions of the corresponding monolignols [1]. The monomeric units syringyl (S), guaiacyl (G), and *p*-hydroxyphenyl (H) (Figure 1) will vary in occurrence based on plant species. Besides the most abundant interconnecting alkyl-aryl ether bond (β-*O*-4), many other types of interconnecting bonds between the monomeric units exist (e.g., α-*O*-4, β-1, β-β, β-5, 5-5, 4-*O*-5; see examples of such linkages in Figure 2g). The native lignin in biomass differs greatly in properties, both physical and chemical, compared to lignin isolated through biomass processing. These types of isolated lignins, called technical or industrial lignins, are available in large volumes from the pulping industry [2], but their properties are very dependent on the isolation method [3] and on the biomass feedstock used [4].

The largest pulping processes—Kraft, sulfite, and soda—are currently utilizing the majority of lignin as fuel, but there are bio-refineries that are aiming to fractionate and utilize all of the biomass, which can lead to better availability of lignin for commercialization. The Lignoboost process [5], which allows the isolation of a lignin fraction from the Kraft process, is one such method but there are many other biomass fractionation technologies such as organosolv, steam explosion, acid hydrolysis, hot water extraction, and ammonia fiber expansion [6]. The different processes will yield different types of lignin, but also the same process with different process parameters [7] could affect the structure and properties of the lignin, therefore it is of great importance to properly characterize technical lignins produced by novel fractionation technologies to optimize their use in different applications, as seen from studies that compared how the chemical structure affects the product, i.e., in carbon fiber [8] and phenol-formaldehyde resin [9]. When studying lignin as a source of fuel or chemicals, the focus has often been to catalytically cleave the alkyl-aryl ether bonds [10,11], and thus for such an end use, it would be preferable to have an intact lignin with large amounts of β-*O*-4 linkages and low degree of condensation. There are also applications when lignin is used in its polymeric form, i.e., used as a copolymer, or in polymer blends, or in composites [12,13,14,15] and in these cases condensation does not necessarily affect the properties negatively. To improve the functional properties of polymer blends, and to increase the compatibility of lignin with other polymers, chemical modification or derivatization is often necessary [16,17,18,19]. The compatibility [20] and the thermal stability [21] can be tailored by altering the derivatization agent e.g., in lignin esters by increasing the length of the esters. The innate properties of polymers can also be improved in blends with lignin as seen when poly lactic acid (PLA) was blended with fatty acid esterified lignin [22]. These blends are often thermally treated or processed, e.g., with temperatures around 170 °C for PLA [23,24] and temperatures up to 220 °C for poly(ethylene oxide) blends [25]. The blending temperature can be crucial as it has been shown that lignin enhance the degradation polymers such as PLA at elevated temperatures [26]. 

In this article we have studied the softwood lignin fraction obtained from a novel pressurized hot water extraction (PHWE) biorefinery process [27,28]. In the first step of the process the hemicelluloses are isolated by extraction from wood chips with hot water (max 150 °C) at oxygen-starved conditions [27]. After the separation of the hemicelluloses the chips are cooked with alkali in a second step to isolate the cellulose fraction as described by Schoultz et al. [28]. The lignin, called BLN lignin, can then be precipitated and purified from the black liquor. The process is currently being up-scaled for commercial applications. In our earlier work we have analyzed the structural changes that the process caused to birch lignin and showed that the process clearly affects the structure [29]. As the process caused condensation and increased the amount of free hydroxyl groups, this lignin may be more suitable for applications where it is used in its polymeric form. The main objective of this study was to investigate the structural and thermal properties of the BLN lignins from the softwoods Norway spruce and Scots pine. The lignins were also chemically modified by acetylation and methylation of the phenolic hydroxyl groups. These modifications were performed as it has been shown that especially free hydroxyl groups are labile to thermal treatments [30]. These types of reactions, or similar types of reactions, are also often used to increase the compatibility with other polymers. The lignin was also fractionated by solvent fractionation, which illustrates that the lignin can be further tailored into more narrow molar mass fractions by simple separation methods. 

## 2. Results and Discussion

### 2.1. Analysis of the Unmodified Lignin 

The structural alteration of the native lignin linkages can be seen from the HSQC spectrum of the MWL (Figure 2a, pine, and 2c, spruce) and BLN lignin (Figure 2b, pine, and Figure 2d, spruce), and the NMR shifts are listed in Table 1. Based on the integrals of the correlation peaks the small amount of remaining β-*O*-4 (A) (From MWL 27 to BLN 5% in pine and from 32 to 4% in spruce), β-β (B) (From MWL 7 to BLN 2% in pine and 5 to 3% in spruce), and β-5 (C) (From MWL 11 to BLN 3% in pine and 15 to 3% in spruce) only constitutes a small fraction compared to the MWL and no dibenzodioxocin (K), cinnamyl alcohol (E), cinnamyl aldehyde (I), or γ-substituted β-*O*-4 (A’) could be detected in the BLN lignins. The aryl glycerol group (J) was formed during the process and is known to be formed during soda pulping [31,32]; however, whether the group is formed during the initial PHWE or during the alkaline pulping was not confirmed. Both enol ethers (L) and stilbenes (M) could be detected in the aromatic region and guaiacyl propanol (F), and secoisolariresinol (D) in the aliphatic region. Two overlapping signals B_γ’_ and J_β_, marked by * in Figure 2b,d, was detected from an unidentified structure. This was deduced as both B_γ’_ and J_β_ show inconsistencies in their integrals from HSQC compared to B_γ_ and Y_α_. The B_γ’_ which should be a CH_2_ in the β-β resinol structure is a CH/CH_3_ in both of the multiplicity edited HSQC lignin spectra, due to the overlapping of signals. From the aldehyde region (Figure 2e,f) we detected the C_γ_-H from the cinnamyl aldehyde (I) moiety and a correlation peak originating from benzylic aldehyde (N) unit in the MWL (Figure 2e), but in the BLN lignin only the benzylic aldehyde (N) correlation peak remains (Figure 2f). There was an increase of carboxylic groups based on ^31^P-NMR (Table 2), however, it seems that not all benzylic aldehydes were oxidized to carboxylic acids during the treatment and this could be due to the oxygen removal during the process. Only small amounts of carbohydrate impurities were detected in the HSQC spectrum of the BLN lignin. It is evident that significant condensation of the lignin occurred as the native lignin linkages were reduced in amount but the lignins retain a large molar mass. The lignins were also enriched with carbon compared to the MWL as seen from the elemental analysis (Table 3). 

The fragmentation and condensation can occur during the slightly acidic PHWE phase and/or during the alkali pulping, the mechanism can be either ionic and/or radical and will most likely occur at the C_α_, C_5_, or C_6_ position. The formation of condensed 5-substituted phenolic structures is seen from the ^31^P-NMR spectrum as they are detected separately from the aliphatic and guaiacylic hydroxyl groups (see Appendix A). The condensation was also seen from the HSQC of the methylated BLN lignin (Figure 3), where only the correlation peak of the methylated 4-OMe in 5-condensed structures was shifted to δ_C_/δ_H_ (60.1/3.72) ppm (Figure 3b,c) and all other aromatic OMe-peaks will remain at approximately δ_C_/δ_H_ (55.5/3.76) ppm. From our previous work on birch lignin we detected evidence of significant 6-C condensation of the aromatic ring, this was detected as the chemical shift of the 5-OMe in the syringylic unit was shifted in a similar manor as the 4-OMe in the 5-condensed methylated guaiacyl unit. We did not see any such shift in the softwoods due to the fact that a condensation on the 6-position in guaiacyl units would not yield an OMe-group which has two neighboring substituents, which leads to the increase in the chemical shift. The broadening and slight shift of the HSQC correlation peaks of C_2_-H δ_C_/δ_H_ (110.3/6.94) to (112.0/6.79) ppm and C_6_-H δ_C_/δ_H_ (118.9/6.83) to (120.3/6.66) ppm compared to that of the MWL is caused by, among other things, the 5-condensation of the guaiacyl units [33]. The integrals of the aromatic C_2_-H, C_5_-H and C_6_-H from the HSQC correlation peaks gave us an indication of the condensation pattern and showed the least amount of C_5_-H followed by C_2_-H and C_6_-H. However, as we have a clear reduction in aliphatic side chains there will most likely also be C_1_-C_α_ cleavage and the chemical shift of the formed C_1_-H is similar to that of C_6_-H, as seen from model compounds [34], which would give an inflated value of the amount of C_6_-H. Other issues with the aromatic signals is the broadening of the signals, and especially as C_2_-H and C_6_-H will be more affected by changes to the side chain compared to C_5_-H but also the increase of free phenolic guaiacyl units will affect the chemical shifts of C_2_-H, C_5_-H and C_6_-H compared to the etherified shifts. Due to the many factors that can affect the chemical shift of the aromatic signal, their integrals should only be considered as semi-quantitative. 

### 2.2. Acetylated and Methylated Lignin

The degree of substitution of the acetylated and methylated lignin was determined by ^31^P-NMR (see Appendix A). The acetylated lignin contained no free hydroxyl groups and the HSQC-spectrum showed characteristic correlation peaks for acetylated structures (see Appendix A). For the analysis of the thermal properties we used Me_2_SO_4_ in alkaline aqueous media to selectively methylate the phenolic hydroxyl groups in the lignin and to leave the aliphatic hydroxyl groups and carboxyl groups unmodifed (see Table 2). The small amounts of remaining free phenolic groups seen in Table 2 seemed, based on ^31^P-NMR shifts of the peaks, to originate from hydroxyl groups in 5-condensed structures. To fully methylate one lignin sample, CH_3_I was used as methylation agent and NaH as base in anhydrous DMF (Figure 3c). This sample was only used to compare the structural differences between the unmodified (Figure 3a), methylated phenolic (Figure 3b), and fully methylated lignin (Figure 3c). The HSQC correlation peaks in the oxygenated aliphatic area can be seen in Figure 3. The selectively methylated lignin (Figure 3b) remained similar to the starting material (Figure 3a), with no changes detected in the structures originating from natively occurring lignin linkages. 

The only difference between Figure 3a,b is the slightly higher chemical shift, in the ^13^C-NMR spectrum, of the 4-OMe in structures that has been condensed on the 5-position. This change in chemical shift can also be seen in model compounds with the same type of structure [34]. In the fully methylated derivative (Figure 3c) all the traditionally occurring C-H correlation peaks, besides the β-β signals that lack hydroxyl groups in the structure, were shifted to a higher ppm value due to the methylation of the aliphatic hydroxyl groups. The methyl ester group and the aliphatic methyl ether groups were also clearly detected separately (Figure 3c). In the aromatic area of the methylated lignin only the C_5_-H correlation peak was shifted to a lower ^13^C shift, in accordance with the fact that it will be more affected by the methylation than the C_2_-H and C_6_-H.

### 2.3. MTBE-Soluble and i-PrOH Fractions

The lignin precipitated from the black liquor was fractionated to MTBE insoluble lignin (BLN) and MTBE soluble lignin (see Scheme 1). The MTBE soluble fraction, ~15 w% of the total amount isolated, contained low molar mass compounds, mainly wood extractives and a complex mixture of phenolic compounds. Some polymeric or oligomeric lignin was also detected from the characteristic signals of the aromatic- and methoxyl groups in the HSQC spectrum (Figure 4). Large amounts of aliphatic, aromatic, and olefinic correlation signals was seen, origination from various lignin degradation products and wood extractives. The HSQC spectrum lacks all the traditional lignin structures as seen from the oxygenated aliphatic area. Large amounts of carboxylic groups and small amounts of aliphatic hydroxyl groups was detected in this fraction based on the ^31^P-NMR analysis (Table 2), which is consistent with wood extractives and lignin degradation products. The MTBE insoluble lignin (BLN lignin) was separated to a high and medium molar mass portion by *i*-PrOH solvent fractionation (see Scheme 1 and Table 3). The method allows us to prepare lignin with a more narrow molar mass. Solvent fractionation is a common method to separate lignin based on size and a wide array of solvents, or solvent mixtures, has been used for this purpose [35,36]. 

Structurally the *i*-PrOH insoluble portion (~70 w%) had a larger molecular mass than the *i*-PrOH soluble portion (~30 w%). It also contained a smaller amount of carboxylic groups and free phenolic G-units compared to the *i*-PrOH soluble fraction. The majority of low molar mass compounds were removed from the starting lignin during the MTBE extraction, but small amounts was still detected in the *i*-PrOH soluble fraction. All of the fractions had a narrower PDI value than the BLN lignin. Low PDI values in lignin are a favorable parameter in producing functional materials as it reduces the complexity of the polymer [19].

### 2.4. Thermal Properties

TGA was used to determine the thermal stability and decomposition of the lignin samples (Figure 5). A birch lignin (Figure 5a) from the same process was used to compare the thermal properties between softwood and hardwood lignin (Table 4). According to the literature the degradation of the lignin structure starts at 230–260 °C with the degradation of the propanoid side chain and then continues to cleave the linkages bonded with C-C bonds at 275–350 °C [37]. The relatively high T_dst 95%_ values of 273–302 °C for the unmodified lignins indicated low amounts of phenylpropanoid side chains and aryl ether linkages and that the lignin was bonded by more stable C-C bonds, which is in agreement with our structural analysis of the lignin. The methylation of the phenolic groups slightly increased the stability of all the lignin samples compared to the unmodified lignin, as seen from the T_dst 95%_ values (Table 4). The acetylation decreased the stability which is contradictory to other studies [38], this indicate that the acetyl groups were cleaved or that the decrease in mass is due to evaporation of less volatile compounds. The cleavage could be catalyzed by some impurity, such as minuscule amounts of acetic acid from the reaction. To study this phenomenon two samples, unmodified pine lignin and acetylated pine lignin, were dried extensively at elevated temperature prior to analysis. The results (see Appendix A) showed an slight increase in the T_dst 95%_ value for the acetylated sample from the initial value of 233 to 264 °C, which can partially be explained by the more efficient removal of volatiles, however, the T_dst 95%_ value was still lower than the 284 °C of the unmodified lignin, which could be due to that the cleaved acetyl groups can act as catalyst for further deacetylation. The acetylated lignins still had high T_dst 95%_ values which make acetylation a viable option to increase the compatibility of lignin with other polymers. The thermal stability can also be increased by performing the same esterification procedure with esters with longer chains. By adjusting the chain of the esters the chemical properties of the lignin can be tailored to increase the compatibility with specific polymers [14]. The use of this lignin in polymer blends could be promising, based on the T_dst 95%_ values, as blend processing temperatures are considerable lower, however, it is clear that the lignin starts to lose mass at lower temperatures and as seen from *Cicala* et al. [26] the processing stability of a lignin/PLA blend was considerably decreased at 190 °C compared to 170 °C. The DTG maxima of the different samples were between 386–411°C with the methylated lignin slightly higher. The unmodified lignin had a higher char residue at 600 °C (wt %) than the methylated lignin, which could be due to the formation of more stable condensed structures in the unmodified lignin with free phenolic groups, while the acetylated samples had a lower char residue. 

## 3. Materials and Methods

### 3.1. Materials

All reagents were purchased from Sigma Aldrich (Espoo, Finland) if not stated otherwise and used without further purification. Milled Wood Lignin (MWL) was prepared according to a previously published procedure [39].

### 3.2. BLN Process

The pressurized hot water extraction process has been described elsewhere [27]. In short, wood chips of Norway spruce (*Picea abies*) or Scots pine (*Pinus sylvestris*) were first extracted with hot water to remove the hemicelluloses, and then the remaining fibers were further cooked with NaOH to give the black liquor that was separated from the pulp [28]. For the thermal analysis also birch lignin (*Betula pendula*) from the same process was used and has been characterized elsewhere [29].

#### 3.2.1. Precipitation, Purification and Fractionation

The lignin was precipitated from the black liquor by addition of 1 M HCl until the pH was 2.5. The lignin was then collected either by careful filtration. The lignin cake was then washed and collected five times with water acidified to pH 2.5 with HCl. After the final wash the lignin slurry was extracted 10 times with MTBE. Both the solid BLN fraction and the MTBE-soluble fraction was dried and analyzed.

#### 3.2.2. iPrOH Fractionation

Purified BLN lignin (MTBE insoluble fraction, 1.0 g) was stirred with iPrOH (40 mL) for 1 h and then centrifuged. After centrifugation the iPrOH was decanted off and the process was repeated 10 times. The iPrOH insoluble (iPrOH insol) and iPrOH soluble (iPrOH sol) fractions were collected separately, dried, and analyzed. 

### 3.3. Elemental Analysis

Elemental analysis was performed on a FLASH 2000 organic elemental analyzer (Thermo Fischer Scientific, Cambridge, UK). No sulphur was detected and the oxygen content was calculated by subtracting the sum of carbon, hydrogen and nitrogen from 100%.

### 3.4. Molar Mass Distribution

Molar-mass characteristics was analysed using a Shimadzu (Shimadzu Corp., Kyoto, Japan) HPLC system (SCL-10AVP system controller + DGU-14A on-line degasser + FCV-10ALVP low-pressure gradient valve + LC-10ATVP HPLC pump + SIL-20AHT autosampler + CTO-10ACVP column oven) equipped with a sequentially connected guard column (50 mm × 7.8 mm) and two Jordi Gel DVB 500A (300 mm × 7.8 mm) columns in series (Columnex LLC, New York, NY, USA). Eluent: THF with 1% acetic acid, flow rate: 0.8 mL/min, column oven temperature 40 °C. Injection volume of the autosampler was 50 μL. Detector: LT-ELSD detector (SEDEX 85 LF Low-Temperature Evaporative Light Scattering Detector, (SEDERE, Alfortville, France). Detector parameters: HPLC nebulizer, 40 °C, air pressure: 3.5 bar, gain 3, no-split mode. Columns calibration was performed using Mono-Disperse Polystyrene Standards (Perkin-Elmer, Norwalk, CT, USA).

### 3.5. NMR Spectroscopy

All the NMR experiments were performed at 298 K in DMSO-d_6_ on an AVANCE III spectrometer (Bruker Biospin GmbH, Rheinstetten, Germany) operating at 500.13 MHz for ^1^H, 125.77 MHz for ^13^C and 202.46 MHz for ^31^P. The ^13^C-NMR were measured with a spectral width of 35,714 Hz, 2 s acquisition and a 10 s relaxation delay with the Bruker pulseprogram *zgig*. HSQC experiments used the Bruker’s pulse program “hsqcedetgpsisp2.3 for multiplicity edited with a spectral width of 8012 Hz (from 0–16 ppm) and 30,182 Hz (from 0–220 ppm) for the ^1^H- and ^13^C-dimensions. A semi quantitative method was used for calculating the amounts of lignin linkages by using the C_2_-H integral as internal standard (IS). A common standard protocol was used for ^31^P NMR sample preparation [40]. To a solution of 20 mg lignin in a 0.4 mL mixture of pyridine and CDCl_3_ (1.6:1, *v/v*) 0.100 mL of a IS solution (0.12 M) was added. After thorough stirring, 0.1 mL of phosphitylation reagent [2-chloro-4,4,5,5-tetramethyl-1,3,2-dioxaphospholane (TMDP)] and 0.050 mL of a Cr(acac)_3_ solution (11.4 mg/mL) was added and the sample was stirred at room temperature before transferred to a NMR tube. The ^31^P NMR measurements were collected with a 2.0 s acquisition time and a 5.0 s relaxation delay. The spectra were calibrated using the signal of the water-derivatized signal at 132.2 ppm. For the MWLs and BLN lignin cyclohexanol was used as internal standard. For the MTBE soluble and IPA fractionated lignin *N*-hydroxy-5-norbornene-2,3-dicarboxylic acid imide (e-HNDI) was used to avoid overlapping of impurities with the IS. 

### 3.6. Thermal Analysis

TGA was carried out using a STA 409 PG/1/G Luxx (NETZSCH-Gerätebau GmbH, Selb, Germany) in the range of the temperature 23 to 600 °C at a rate of 10 °C/min under N_2_ atmosphere. As the different samples: unmodified lignin, acetylated lignin, and lignin with methylated phenolic groups had different workup procedures prior to drying under vacuum, the 100 wt% was set to the value when the samples had been heated to 120 °C during the analysis. Any prior weight losses correspond to the loss of moisture or volatile compounds, and it was concluded that at ~120 °C the weight loss had stabilized based on DTG. To study the possible deacetylation two samples, unmodified pine lignin and acetylated pine lignin, was dried under vacuum at 60 °C over one week to ensure removal of any traces of acetic acid and then analyzed by TGA. The samples were also preheated to 100 °C and cooled prior to analysis to ensure removal of moisture that could have been absorbed on the lignin during transfer to the instrument.

### 3.7. Acetylation

Lignin (100 mg) was dissolved in pyridine (1.0 mL) and acetic anhydride (1.0 mL) was added. The mixture was stirred in darkness for 3 days before the reaction mixture was cooled and quenched by addition of MeOH and evaporated under reduced pressure. The crude product was redissolved in CHCl_3_, extracted three times with 0.1 M HCl, twice with water, dried with Na_2_SO_4_, and finally concentrated under vacuum with isolated yields over 90%. 

### 3.8. Methylation

A previously reported method for methylation of the phenolic hydroxyl groups was used [41]. In short, the lignin (1.0 g) was dissolved in 0.7 M NaOH (15 mL) and Me_2_SO_4_ (0.95 mL, 10.0 mmol) was added. The mixture was stirred for 30 min at room temperature followed by 2 h at 80 °C while 0.7 M NaOH was continuously added to keep the solution alkaline according to the procedure. The amounts of Me_2_SO_4_ was calculated from the total amount of free phenolic groups from the ^31^P NMR analysis, approximately 2.5–3.0 equivalents of Me_2_SO_4_ per phenolic hydroxyl group was used in this work.

For the complete methylation the lignin (2.0 g) was dissolved in dry DMF (30 mL). An appropriate amount of 60% NaH in mineral oil (5 equivalents with respect to the total amount of free hydroxyl groups) was measured and washed three times with hexane. The NaH was then stirred to a suspension with 10 mL dry DMF and added dropwise to the lignin solution on ice bath. The methylation agent CH_3_I (5 equivalents) was then added dropwise. After 16 h the reaction was cooled on an ice bath and the excess NaH was quenched by addition of MeOH. The solution was then poured into a large volume of water and acidified to pH 2.5 with 1M HCl. The lignin was collected by filtration and was purified twice by stirring the mixture in acidified water (pH 2.5) and filtering in between. After the final filtration the cake was thoroughly washed with distilled water and freeze dried prior to analysis.

## 4. Conclusions

In this study we have determined the structural characteristics and thermal properties of softwood lignins obtained from a novel PHWE process and their acetylated and methylated derivatives. It was found that the lignin from the process, which is optimized for obtaining the carbohydrate fraction from wood, clearly altered the lignin structure when compared to MWL from the same wood chips. The sulphur-free lignin shared many common traits shared among technical lignin [42] such as the reduction of the aliphatic chains and natively occurring lignin linkages. Condensed structures that had formed during the process were detected, mainly at the C_5_-H position of the aromatic ring. The polymeric lignin had larger amounts of carboxylic groups and free phenolic groups compared to MWL. The lignin from the two different softwood species showed very little differences, both structurally and in thermal properties, and can as such be used as mixtures for potential applications. The thermal properties of the softwood lignins were compared to hardwood birch lignin from the same process and the properties were similar, for both unmodified and modified lignin. It was also shown that the BLN lignin could easily be separated by solvent fractionation to prepare narrower molar mass portions with slightly different functional group compositions. The lignins had high T_dst 95%_ values (273–302 °C) and the free phenolic groups could easily be chemically modified to tailor the chemical properties, which make this lignin and its derivatives promising candidates for the use in polymer blends.

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
