# Peer review of "Structural and Thermal Analysis of Softwood Lignins from a Pressurized Hot Water Extraction Biorefinery Process and Modified Derivatives"

_molecules, 2019, doi:10.3390/molecules24020335_

Reviewer 1 Report

Manuscript is well written and provided interesting technological data, which might support some lignin applications. 

However, the following topics must be improved:

Abstract

The following sentence lacks of conciseness (highly speculative), once thermal stability cannot define (alone) if a material is suitable to be applied in polymeric blends, composite and resins. This sentence must be rewritten.

"The lignin was determined to be thermally stable up to (273-302 °C) based on its Tdst 95% value. As such it was concluded that it is a suitable to be used in polymeric blends, composites or in resins."

Introduction

The introduction does not present clearly what is the "main problem" treated in this work. In this current form, the manuscript seems to be an industrial report, in which the structural and thermal properties of several "types of lignin" are simply presented.

Authors have to elucidate how these properties or differences could affect or support lignin applications, in polymeric system (for instance), as proposed by them.

Note, they claim that these materials are suitable for "polymer developments", on the other hand, they cited an unique reference [12], which relates the influence of some lignin strucutures on polymer properties. I recognize the relevance of this reference and its authors [12], but there are many other results publications, which can be cited to stregthen these arguments.

Please revise the introduction taking in consideration the comments above. Number of reference must be increased in the introduction.

Discussion

        2.4 Thermal properties

         The following sentences must be presented into the Materials and Methods section:       

        TGA was used to determine the thermal stability and decomposition of the lignin samples (Figure 5). A birch lignin (Figure 5a) from the same process was used to compare the thermal properties between softwood and hardwood lignin. As the different samples: unmodified lignin, acetylated lignin, and lignin with methylated phenolic groups had different workup procedures, the 100 wt% was set to the value when the samples had been heated to 120 °C during the analysis.....

        To study this phenomena two samples,unmodified pine lignin and acetylated pine lignin, was dried under vacuum at 60 °C over one week and then analyzed by TGA. The samples were also preheated to 100 °C and cooled prior to analysis to ensure removal of moisture that could have been absorbed on the lignin during transfer to the instrument.

    In general, the discussion regarding thermal properties is poor. The lignins clearly started to loss mass at temperatures lower than (273-302 °C), which might affect processing and polymer degradation, for example, if these lignin are applied on polymer developments. 

     Once again the authors used an unique reference to support their complete discussion, regarding thermal properties.      

Author Response

The following sentence lacks of conciseness (highly speculative), once thermal stability cannot define (alone) if a material is suitable to be applied in polymeric blends, composite and resins. This sentence must be rewritten.

"The lignin was determined to be thermally stable up to (273-302 °C) based on its Tdst 95% value. As such it was concluded that it is a suitable to be used in polymeric blends, composites or in resins."

The abstract has been partially rewritten (lines 19-24): In addition to the thermal stability we included that the narrow PDI and the possibility to modify the lignin should be beneficial for the said application.

Introduction

The introduction does not present clearly what is the "main problem" treated in this work. In this current form, the manuscript seems to be an industrial report, in which the structural and thermal properties of several "types of lignin" are simply presented.

Authors have to elucidate how these properties or differences could affect or support lignin applications, in polymeric system (for instance), as proposed by them.

Note, they claim that these materials are suitable for "polymer developments", on the other hand, they cited an unique reference [12], which relates the influence of some lignin strucutures on polymer properties. I recognize the relevance of this reference and its authors [12], but there are many other results publications, which can be cited to stregthen these arguments.

Please revise the introduction taking in consideration the comments above. Number of reference must be increased in the introduction.

The introduction  has been rewritten.We have increased the discussion regarding lignin-polymer blends. Specifically how chemical modifications are often necessary to increase the compatibility with other polymers. We also discuss how these modifications can affect the blends and we described the thermal treatments or processing often used for example with PLA/lignin blends. The number of references was increased. Ref. 13-17 and 19-26 were added and ref. 18 was moved.

Discussion

        2.4 Thermal properties

         The following sentences must be presented into the Materials and Methods section:

     TGA was used to determine the thermal stability and decomposition of the lignin samples (Figure 5). A birch lignin (Figure 5a) from the same process was used to compare the thermal properties between softwood and hardwood lignin. As the different samples: unmodified lignin, acetylated lignin, and lignin with methylated phenolic groups had different workup procedures, the 100 wt% was set to the value when the samples had been heated to 120 °C during the analysis.....

        To study this phenomena two samples,unmodified pine lignin and acetylated pine lignin, was dried under vacuum at 60 °C over one week and then analyzed by TGA. The samples were also preheated to 100 °C and cooled prior to analysis to ensure removal of moisture that could have been absorbed on the lignin during transfer to the instrument.

In results and discussion: The above mentioned text was moved into material and methods section 3.6. The text was slightly altered to increase the clarity of the text.

    In general, the discussion regarding thermal properties is poor. The lignins clearly started to loss mass at temperatures lower than (273-302 °C), which might affect processing and polymer degradation, for example, if these lignin are applied on polymer developments. 

Under results and discussion , thermal properties: We have added a more common discussion on lignin degradation (207-211). On lines 224-230 we increased the discussion regarding chemical modification to increase the stability and compatibility with other polymers. We also added a discussion of how the lignin starts to degrade at an earlier temperature which could affect chemical blends.

     Once again the authors used an unique reference to support their complete discussion, regarding thermal properties.

A more detailed discussion was both added in the Introduction and in Results and discussion. The added discussion in the introduction was based on, besides the previous references, 14 new additional references (13-26).

In addition, several small corrections and improvments have been conducted, based on the reviewers comments.

Reviewer 2 Report

Lagerquist et al studied the lignins and their derivatives after PHWE, and accumulated some interesting data for characterization. Below are my comments, hope they are helpful for the authors. My recommendation of this work is major revision.

1.       In introduction, the authors might have to add a figure depicting the chemical structures for background information.

2.       Figure 3. There is an extra line between b and c.

3.       Figure 1 in page 3 needs to be integrated with figure 1 e. Together with figure 2, the ppm is partially covered due to overlapping. Please modify.

4.       Supporting information. The authors needs to provide full range nmr spectra in addition to the ones with partial x and y axis.

5.       Figure s5. There is no ppm information as x axis.

6.       Figure s12. There is no units for x and y axis.

7.       Line 20. The i- in isopropanol should be italic. Check other places.

8.       Line 36. -O-4, the O should be italic. Check other places.

9.       Line 24. Remove a.

10.   Table 3. Use () instead of [].

Author Response

1.       In introduction, the authors might have to add a figure depicting the chemical structures for background information.

Page 2. Figure 1 (new figure) showing the lignin monomeric structures was added, and referred to in the text line 38. After the discussion of different lignin linkages the reader is referred to figure 2g (figure of the different lignin linkages).

2.       Figure 3. There is an extra line between b and c.

A. page 9. The pictures in figure 3 was revised to remove the extra line.

3.       Figure 1 in page 3 needs to be integrated with figure 1 e. Together with figure 2, the ppm is partially covered due to overlapping. Please modify.

The HSQC figure (old figure 2) of the aldehyde peaks was removed and integrated as part of Figure 2 (old figure 1) as figure 2e and 2f. The structures in the figure was also removed and added into the figure 2g. All figures in figure 2 were resized to avoid the overlapping that caused the partial coverage of the ppm-scale.

4.       Supporting information. The authors needs to provide full range nmr spectra in addition to the ones with partial x and y axis.

A. We have added full range HSQC spectra of the methylated lignin into the supporting info (S4-S7). The HSQC of the acetylated spectra had their ppm range increased (S1-S3)

5.       Figure s5. There is no ppm information as x axis.

A. Added ppm scale on x axis on S9 (old S5)

6.       Figure s12. There is no units for x and y axis.

A. Added units for x and y axis on S16 (old S12)

7.       Line 20. The i- in isopropanol should be italic. Check other places.

A. Corrected

8.       Line 36. -O-4, the O should be italic. Check other places.

A. Corrected

9.       Line 24. Remove a.

A Corrected

10.   Table 3. Use () instead of [].

A: Corrected

In addition, several small corrections and improvments have been conducted, based on the reviewers comments.

Round  2

Reviewer 2 Report

thanks for the revision, i feel comfortable to recommend acceptance of this work.